# Comparative Proteomics Reveals the Difference in Root Cold Resistance between *Vitis. riparia* × *V. labrusca* and Cabernet Sauvignon in Response to Freezing Temperature

**DOI:** 10.3390/plants11070971

**Published:** 2022-04-02

**Authors:** Sijin Chen, Hongyan Su, Hua Xing, Juan Mao, Ping Sun, Mengfei Li

**Affiliations:** 1State Key Laboratory of Aridland Crop Science, Gansu Agricultural University, Lanzhou 730070, China; chenjj@gsau.edu.cn (S.C.); Shy92232@163.com (H.S.); xingh@gsau.edu.cn (H.X.); 2College of Horticulture, Gansu Agricultural University, Lanzhou 730070, China; maojuan@gsau.edu.cn

**Keywords:** grapevine, root cold resistance, *V**itis. riparia* × *V**itis. labrusca*, Cabernet Sauvignon, proteomics, gene expression

## Abstract

Grapevines, bearing fruit containing large amounts of bioactive metabolites that offer health benefits, are widely cultivated around the world. However, the cold damage incurred when grown outside in extremely low temperatures during the overwintering stage limits the expansion of production. Although the morphological, biochemical, and molecular levels in different *Vitis* species exposed to different temperatures have been investigated, differential expression of proteins in roots is still limited. Here, the roots of cold-resistant (*Vitis. riparia* × *V. labrusca*, T1) and cold-sensitive varieties (Cabernet Sauvignon, T3) at −4 °C, and also at −15 °C for the former (T2), were measured by iTRAQ-based proteomic analysis. Expression levels of genes encoding candidate proteins were validated by qRT-PCR, and the root activities during different treatments were determined using a triphenyl tetrazolium chloride method. The results show that the root activity of the cold-resistant variety was greater than that of the cold-sensitive variety, and it declined with the decrease in temperature. A total of 25 proteins were differentially co-expressed in T2 vs. T1 and T1 vs. T3, and these proteins were involved in stress response, bio-signaling, metabolism, energy, and translation. The relative expression levels of the 13 selected genes were consistent with their fold-change values of proteins. The signature translation patterns for the roots during spatio-temporal treatments of different varieties at different temperatures provide insight into the differential mechanisms of cold resistance of grapevine.

## 1. Introduction

The fruits of the grapevine (*Vitis vinifera* L.) are rich in phenolics, flavonoids, and resveratrol with many biological activities such as antioxidants, cardiovascular benefits, and cancer chemopreventive activity [1,2]. Nowadays, grapevines are cultivated in many countries around the world, principally distributed in Europe [3]. In China, the optimal regions for cultivation are mainly distributed in Gansu, Inner Mongolia, Ningxia, Shaanxi, Shandong, and Xinjiang [4]. Currently, the 13 varieties of *Vitis* that are cultivated on a large scale (>100,000 hm^2^) around the world mainly include Cabernet Sauvignon, Merlot, Chardonnay, Syrah, Sauvignon Blanc, and Pinot Noir [5].

In commercial large-scale cultivation, grapevines are frequently exposed to environmental stresses such as drought, salinity, and extreme temperatures [6]. Most cultivated grapevines are suited to grow in temperate and subtropical regions with mild winter conditions [7]. However, grapevines are often grown outside with severe winter conditions characterized by low temperatures, which limits the current and future expansion of production [8]. In order to diminish the freeze damage, several efforts including evaluating cold-resistant species or varieties, as well as revealing the mechanisms of cold resistance, have been undertaken. Evaluating the cold-resistant species or varieties, *V. riparia* is the most cold-resistant species and *V. labrusca* belongs to medium resistance among seven wild *Vitis* species native to North America [9]; the cold resistance of *V. riparia* × *V. labrusca* (Beta) and *V. berlandieri* × *V. riparia* (5BB) is greater than that of Cabernet Sauvignon and Merlot [10]. Regarding the mechanisms of cold resistance, physiological and biochemical metabolites (e.g., soluble sugars, proteins, and hormones) in buds, branches, and roots were determined [11,12,13]; key genes (e.g., *CBF/DREB*, *ICE,* and *AP2/ERF*) enhancing freezing tolerance were identified [14,15,16]; and differentially expressed genes among different species and temperatures were analyzed by transcriptomics [8,17,18]. 

Previous studies have reported that grape branches and buds can survive temperatures of −13 °C or lower, but roots have weaker cold resistance than the above-ground parts [19,20]. To date, although the levels of osmoregulatory metabolites, the activities of antioxidant enzymes, and the expression levels of cold resistance genes between different *Vitis* species in low temperatures have been investigated in extensive experiments [11,12,13,14,15,16,17,18], the differential expression of proteins in roots between cold-resistant and cold-sensitive species or varieties in different low temperatures has not been determined or identified. In this study, the roots of cold-resistant (*V. riparia* × *V. labrusca*, T1) and cold-sensitive varieties (Cabernet Sauvignon, T3) at −4 °C and also at −15 °C for the former (T2) after 30 d of T1 were spatio-temporally measured by quantitative iTRAQ-based proteomic analysis. We found that 25 proteins were differentially co-expressed during the three treatments and their biological functions were involved in stress response, bio-signaling, metabolism, energy, and translation; the expression levels of related genes were validated by qRT-PCR.

## 2. Results

### 2.1. Difference in Root Activity 

After the roots measured by the triphenyl tetrazolium chloride (TTC) method, the root activity was evaluated based on the amount of reactant triphenylformazan. As shown in Figure 1, there was a 1.68-fold decrease in root activity for T2 compared to T1, and T1 root activity was 2.54-fold times greater than that of T3. Moreover, the root activities during the different treatments suggest that the roots can be used for proteomic analysis.

### 2.2. Analysis of Differentially Expressed Proteins (DEPs) 

To reveal the cold resistance of root in grapevine, the DEPs in roots of *V. riparia* × *V. labrusca* and Cabernet Sauvignon for different treatments were analyzed by iTRAQ. A total of 36 and 57 DEPs were obtained in T2 vs. T1 and T1 vs. T3, respectively (Figure 2; Appendix A). The heatmaps and clustering of the DEPs in T1, T2, and T3 are shown in Figure 3, indicating that the quality of proteins was well controlled, and the data of differential expression could be further analyzed. 

The functions of the DEPs were compared to the GO database (Figure 4). In T2 vs. T1, 32 proteins were sorted into biological processes, including cellular macromolecule metabolic process (14, GO: 0044260), cellular protein metabolic process (13, GO: 0044267), proteolysis involved in cellular protein catabolic process (3, GO: 0051603), iron ion transport (1, GO: 0006826), and cellular iron ion homeostasis (1, GO: 0006879). Moreover, 31 proteins were sorted into molecular functions, including catalytic activity (26, GO: 0003824), phosphotransferase activity, alcohol group as acceptor (4, GO: 0016773), and ATP:ADP antiporter activity (1, GO: 0005471) (Figure 4A). In T1 vs. T3, 56 proteins were sorted into biological processes, including response to stimulus (15, GO: 0050896), response to stress (14, GO: 0006950), response to oxidative stress (6, GO: 0006979), response to biotic stimulus (5, GO: 0009607), defense response (5, GO: 0006952), cation transport (5, GO: 0006812), proteasome-mediated ubiquitin-dependent protein catabolic process (2, GO: 0043161), ATP hydrolysis coupled proton transport (2, GO: 0015991), and metal ion transport (2, GO: 0030001). Furthermore, 13 proteins were sorted into cellular components, including nucleosome (7, GO: 0000786) and thylakoid (6, GO: 0009579), and 30 proteins were sorted into molecular functions, including peroxidase activity (6, GO: 0004601), calcium ion binding (6, GO: 0005509), heme binding (5, GO: 0020037), ion transmembrane transporter activity (4, GO: 0015075), hydrogen ion transmembrane transporter activity (3, GO: 0015078), NADH dehydrogenase (ubiquinone) activity (2, GO: 0008137), triose-phosphate isomerase activity (2, GO: 0004807), and O-methyltransferase activity (2, GO: 0008171) (Figure 4B).

Among the 36 and 57 DEPs in T2 vs. T1 and T1 vs. T3, a total of 25 proteins were differentially co-expressed (Figure 5). Based on the biological functions annotated against the SwissProt, the 25 DEPs were classified into five categories: stress response (6), bio-signaling (4), metabolism (6), energy (6), and translation (3) (Figure 5; Table 1).

### 2.3. Expression Levels of DEPs in Response to Low Temperatures between the Two Varieties

As is shown in Table 1, the six DEPs involved in stress response (DHN1, SHSPCP, USPCP, FER, GluDP, and GPX) showed 1.28–2.83-fold UR in T2 vs. T1 and 1.73–2.69-fold UR in T1 vs. T3. The four DEPs involved in bio-signaling (PKCP, S/TPP, nsS/TPK, and RAD23) showed 1.22–1.98-fold UR in T2 vs. T1 and 1.27–2.92-fold UR in T1 vs. T3. The six DEPs involved in metabolism (GluP, GluBE, PE, ABHD3CP, ProIP, and MT) were differentially expressed in T2 vs. T1, with 2.65-, 2.28-, and 2.61-fold UR for GluP, ProIP, and MT and 0.58-, 0.63-, and 0.84-fold DR for GluBE, PE, and ABHD3CP, respectively. In T1 vs. T3, these six proteins showed 1.21–2.92-fold UR. The six DEPs involved in energy (AAC, AAACP, NADCP, NDUFB7, PCP, and SDHFS) showed 1.59–1.85-fold UR in T2 vs. T1 and 1.59–2.83-fold UR in T1 vs. T3. The three DEPs involved in translation (rpL14, rpS21, and PPI) showed 1.61–2.76-fold UR in T2 vs. T1 and 1.67–2.80-fold UR in T1 vs. T3.

### 2.4. Relative Expression Levels (RELs) of Genes in Response to Low Temperatures between the Two Varieties

In order to validate the expression levels of DEPs, the RELs of genes that accordingly encode DEPs were determined by qRT-PCR. As is shown in Figure 6, the RELs of representative genes were almost consistent with the expression levels of their encoded DEPs. Specifically, the RELs of the stress response genes *DHN1*, *FER,* and *GPX* showed 2.77-, 4.83-, and 3.62-fold UR in T2 vs. T1 and 3.28-, 3.74-, and 2.27-fold UR in T1 vs. T3, respectively (Figure 6A). The RELs of the bio-signaling genes *S/TPP* and *RAD23* showed 3.88- and 2.42-fold UR in T2 vs. T1 and 1.41- and 2.61-fold UR in T1 vs. T3, respectively (Figure 6B). The RELs of the metabolism genes *GluP*, *PE,* and *MT* showed differential regulation with 4.61-, 0.89-, and 4.85-fold in T2 vs. T1 and 2.38-, 1.86-, and 2.63-fold UR in T1 vs. T3, respectively (Figure 6C). The RELs of the energy genes *AAC*, *NADCP,* and *PCP* showed 3.85-, 3.43-, and 3.63-fold UR in T2 vs. T1 and 1.67-, 2.14-, and 4.78-fold UR in T1 vs. T3, respectively (Figure 6D). The RELs of the translation genes *rpL14* and *PPI* showed 3.52- and 3.45-fold UR in T2 vs. T1 and 1.28- and 1.21-fold UR in T1 vs. T3, respectively (Figure 6E).

## 3. Discussion

Plants are frequently exposed to environmental stresses such as low temperature, which plays a major role in the distribution of plant species. Adaptation and acclimation to cold stress result from integrated events occurring at all levels of organization, from the anatomical and morphological level to the cellular (e.g., changes in cell cycle, division, and wall architecture), biochemical (e.g., producing osmoregulatory compounds such as proline and glycine betaine), and molecular levels (e.g., linking the perception of a stress signal with the genomic responses) [21]. In this study, we found that there was a greater root activity in *V. riparia* × *V. labrusca* than Cabernet Sauvignon, and the root activity declined with the decrease in temperature. A total of 25 proteins were differentially co-expressed in *V. riparia* × *V. labrusca* and Cabernet Sauvignon at −4 °C and/or −15 °C treatments, and these 25 DEPs were classified into five categories: stress response, bio-signaling, metabolism, energy, and translation. Meanwhile, the upregulation of the DEPs was observed in cold-resistant *V. riparia × V. labrusca* compared with cold-sensitive Cabernet Sauvignon.

The root system is an important organ absorbing water and minerals from the soil, storing foods (e.g., starch, polysaccharides, and secondary metabolites) and synthesizing the vital substances (e.g., amino acids, hormones, and vitamins) for plant growth and development [22,23]. Root activity is a physiological index that can directly reflect plant growth, nutritional status, and yield level. Here, a greater root activity at −4 °C was observed in *V. riparia* × *V. labrusca* than in Cabernet Sauvignon, affirming that the cold resistance of *V. riparia* × *V. labrusca* is higher than that of Cabernet Sauvignon in the field [10], which also shows that it is of great importance to confer biotic and abiotic stress resistance by grafting scions onto rootstocks. 

DEPs related to stress response may have critical roles in enhancing the cold resistance. Previous studies have reported that dehydrin (DHN) are highly hydrophilic proteins that are involved in cold acclimation processes [24]. Small heat shock proteins (sHSPs) are ubiquitous stress proteins proposed to act as chaperones and have been ascribed an unusual diversity of functions in the cellular response to environmental stress [25]. Universal stress proteins (USPs) are stress-responsive proteins that may contain a single USP domain or two tandem repeats of USP domains [26,27]. Ferritins (FERs) are a broad superfamily of iron storage proteins; exert a fine tuning of the quantity of metal required for metabolic purposes and help plants to protect against oxidative stress [28]. The GluDP plays a role in cell protection against oxidative stress by detoxifying peroxides [29]. Glutathione peroxidases (GPXs) are key enzymes of the cell antioxidant defense system and are involved in scavenging oxyradicals [30]. Investigations have found that the genes *DHN* in tomato [31], *shsp16.9* in rice [32], *USPs* in *Arabidopsis* [33], *TaFER-5B* in *Triticum aestivum* [34], and *GluPX* in *Taxus chinensis* [35] were upregulated in response to cold stress. In this study, the three selected genes *DHN1*, *FER,* and *GPX* were upregulated in T2 vs. T1 (−15 °C vs. −4 °C for *V. riparia × V. labrusca*) and in T1 vs. T3 (*V. riparia × V. labrusca* vs. Cabernet Sauvignon at −4 °C), consistent with the expression levels of their encoded DEPs. The upregulation of these proteins will provide multiple biological functions to coordinate their relationship with low temperature.

Up- or downregulated DEPs related to bio-signaling can perceive the cold stress and transfer it to the cellular response. Previous research has reported that protein kinases (PKs) and protein phosphatases (PPs) play important roles in determining the magnitude and duration of a signaling event, with PKs catalyzing the transfer of a phosphate moiety from ATP to proteins and PPs acting to remove this phosphate group by hydrolysis [36]. The protein serine/threonine phosphatases family from plants constitute PP1, PP2A, PP2B, and novel phosphatases, which have multiple biological functions by regulating a wide variety of cellular signal transduction pathways in response to stresses [37]. The RAD23 is involved in cell cycle regulation, protein quality control, DNA damage response, and cellular metabolism [38]. Investigations have found that the genes *GsLRPK* in *Glycine soja* [39], *S/TPP* in rice [40], and *RAD23* in apple [41] were over-expressed in response to cold stress. In this study, the two selected genes *S/TPP* and *RAD23* were upregulated in T2 vs. T1 and in T1 vs. T3, consistent with the expression levels of their encoded DEPs. The upregulation of these proteins may be constantly on the alert to ensure that plants are not injured by low temperatures.

DEPs related to metabolism can produce osmolytes to protect the cells from cold stress. Previous research has reported that GluP is an important allosteric enzyme in carbohydrate metabolism [42]. The GluBE is involved in the pathway starch biosynthesis, which is part of glycan biosynthesis [43]. The PE is involved in the pathway pectin degradation and in glycan metabolism [44]. The ABHD family of proteins in plants influences the lipid biosynthesis more towards leaf lipids, such as galactolipids, and less towards storage lipids [45]. The ProIP specifically catalyzes hydrolysis of N-terminal proline from peptides [46]. The MT is involved in the sterol and steroid biosynthesis [47]. Extensive experiments have demonstrated over-expression and activity of enzymes that participate in soluble sugar biosynthesis and starch degradation produce proper metabolites to adjust the metabolism and physiology of the plant to cold stress [48]. In this study, the three selected genes *GluP*, *PE*, and *MT* were upregulated in T2 vs. T1 and in T1 vs. T3, which were also consistent with the expression levels of their encoded DEPs. Here, the upregulation of the proteins (GluP, GluBE, PE, ABHD3CP, ProIP, and MT) was observed in T1 vs. T3, while the downregulation for the proteins (GluBE, PE, and ABHD3CP) in T2 vs. T1. The downregulation at −15 °C might be a part of the mechanism associated with the delay of senescence and death [49]. 

DEPs related to energy can accelerate the electron transport rate to defend against low temperatures. Previous studies have reported that AAC plays a key role in the energetic cell metabolism because it exchanges ATP and ADP, the product and substrate of the mitochondrial ATP synthase, respectively [50]. The AAA protein family is a group of ATPases that are associated with various cellular activities [51]. The NAD(P) plays a crucial role in pro-oxidant and antioxidant metabolism and the NAD contents are both flexible and potentially important in determining cell fate [52]. The NDUFB7 is an accessory subunit of the mitochondrial membrane respiratory chain NADH dehydrogenase (Complex I), which functions in the transfer of electrons from NADH to the respiratory chain [53]. The phytocyanins (PCs) are a class of plant-specific blue copper proteins and play critical roles in plant growth and development [54]. The SDHFS is involved in complex II of the mitochondrial electron transport chain and is responsible for transferring electrons from succinate to ubiquinone [55]. Extensive experiments have demonstrated that the membranes become less fluid and the protein components can no longer function normally in cold-sensitive plants, resulting in inhibition of H^+^-ATPase activity, energy transduction, and enzyme-dependent metabolism [21]. In this study, the three selected genes *AAC*, *NADCP*, and *PCP* were upregulated in T2 vs. T1 and in T1 vs. T3, consistent with the expression levels of their encoded DEPs. The upregulation of these proteins will provide energy for the root to maintain activity and growth in response to low temperatures.

DEPs related to translation may be required for maintaining the activation of translation in response to cold stress. Previous works have reported that the large and small subunits of ribosomal proteins are structural constituents of ribosomes, which perform the essential task of protein synthesis in the cell [56]. The PPI functions in the folding of membranal proteins [57]. Investigations have found that the genes *SOL34* in *Glycine max* [58], *RPS5* in *Arabidopsis* [59], and *OsCYP19-4* with PPI activity in rice [60] were over-expressed in response to cold stress. In this study, the two selected genes *rpL14* and *PPI* were upregulated in T2 vs. T1 and in T1 vs. T3, consistent with the expression levels of their encoded DEPs. The over-expression of these proteins could be required for growth acclimation to cold stress during the overwintering stage. 

## 4. Materials and Methods

### 4.1. Plant Material 

The one-year-old seedlings (own-rooted by cutting propagation) of *V. riparia* × *V. labrusca* (cold-resistant variety) and Cabernet Sauvignon (cold-sensitive variety) were planted and grown in a field in Yuzhong County, Gansu, China (1580 m a.s.l.; 35°46′17″ N,104°0′36″ E). Glasses (depth 50 cm, width 150 cm) were used to separate the roots from each other (Appendix A). Complex fertilizer (N + P_2_O_5_ + K_2_O ≧ 500 g/L, Cu + Fe + Mn + Zn + B: 3–30 g/L; 100 mL per plant) purchased from a company (Germany Mike Reze Agricultural Co., Ltd., Stuttgart, Germany) was applied each year in the sandy soil at the depth from 10 to 30 cm, and the soil water content was controlled from 45% to 55% by mulching film, monitored using a rapid soil moisture tester (SFY-100, Shenzhen Guanya, Shenzhen, China). 

After three years, the lateral roots of *V. riparia* × *V. labrusca* were collected at the depth of 20 cm when the average temperatures of the soil surface were −4 °C (T1; 12 December) and −15 °C (T2; 12 January), respectively. The lateral roots of *V. vinifera* were collected at −4 °C (T3; 12 December). The data of 20 cm soil temperatures from December to January at the experiment site were shown in Appendix A. During the collection of roots, the freezing sandy soil containing the roots was first dug out with a rigid shovel, then immediately placed in liquid nitrogen to break down its granular structure. Finally, the flexible lignified roots were picked up and frozen in liquid nitrogen for the measurement of root activity and analysis of proteomics. Each treatment for T1–T3 had nine biological repeats (nine plants).

### 4.2. Measurement of Root Activity 

Root activity was measured according to a triphenyl tetrazolium chloride (TTC) method [61] with slight modifications. Briefly, root-tip samples (100 mg) were cut into pieces and then placed into a glass tube (10 mL). The TTC solution (0.4% *w*/*v*, 3 mL) and Na_2_HPO_4_-KH_2_PO_4_ buffer (0.1 mol/L, 3 mL, pH 7.0) were sequentially added to the tube. After incubating at 37 °C for 1 h, H_2_SO_4_ (1 mol/L, 1.5 mL) was added to the mixture to stop the reaction. The colored samples were transferred to a sealed tube, methanol (15 mL) was added, and then the mixture was incubated at 37 °C for 4 h to decolor. Absorbance readings were taken at 485 nm, and root activity was evaluated based on micrograms of triphenylformazan.

### 4.3. Protein Extraction, Quantification, and Digestion 

Total protein samples were extracted according to previous protocol [62], with some modifications. Briefly, root samples (0.5 g) were ground into powder in liquid nitrogen and dissolved in phosphate buffer (pH 7.4, 20 mL), then added in pre-chilled (−20 °C) trichloroacetic acid/acetone (10% *w*/*v*, 5 mL). After precipitating at −20 °C for 12 h, the homogenate was centrifuged at 14,000× *g* at 4 °C for 15 min. After the supernatant was removed and the precipitate was suspended in acetone at −20 °C for 2 h, the suspension was centrifuged at 14,000× *g* at 4 °C for 10 min. Following exhaustive suspension in acetone (×3), the precipitate was dissolved in triethylammonium bicarbonate (0.5 mol/L, 0.5 mL) at 4 °C for 1 h and then centrifuged at 14,000× *g* at 4 °C for 10 min. Finally, the supernatant was transferred to a new tube. The quality of the extracted protein was examined by SDS-PAGE (Appendix A). The extracted protein was quantified by a Bradford assay using bovine serum albumin as the standard [63]. After protein quantification, an equal amount of proteins (150 μg) were digested using a filter-aided sample preparation method [64].

### 4.4. iTRAQ Labeling and Strong Cation Exchange (SCX) Chromatography Fractionation

The digested peptides were labeled using iTRAQ reagents (iTRAQ^®^ Reagents-8plex kit, Sigma Chemical Co., St. Louis, MO, USA) [65,66,67]. The labeled samples were pooled and purified using SCX chromatography on an Agilent 1260 HPLC (Agilent Technologies Inc., Santa Clara, CA, USA) [68].

### 4.5. Liquid Chromatography (LC)-Electrospray Ionization (ESI) Tandem MS/MS Analysis

LC-MS/MS was performed with an Easy nLC (Thermo Fisher Scientific, Waltham, MA, USA) coupled to Q Exactive MS (Thermo Finnigan, San Jose, CA, USA) [69]. Briefly, the iTRAQ-labeled peptides (5 μg) were separated by a Thermo Scientific Easy C_18_ column (75 μm × 100 mm, 3 μm) with gradient elution from 2% B to 45% B in 120 min (A: 0.1% formic acid in H_2_O; B: 0.1% formic acid in acetonitrile) at a flow rate of 300 nL/min. All tandem MS were produced following the higher collision energy dissociation (HCD) method. Specifically, MS survey scans were acquired using a data-dependent top 10 method, in which the most abundant precursor ions between 350 and 1500 *m*/*z* were dynamically chosen for higher collision energy dissociation (HCD) fragmentation. The resolution was set to 60,000 at 400 *m*/*z*, the automatic gain control (AGC) target value was 1 × 10^6^, and the maximum ion accumulation time was 200 ms. 

### 4.6. Protein Identification and Function Annotation

Protein identification was performed using a decoy database search with the false discovery rate of <1.0% and more than one identified peptide. Protein quantitation was analyzed using an iTRAQ 8-plex combined with the Mann–Whitney test. A criterion of |log_2_(fold-change)| ≥ 1 with a *p*-value of ≤0.05 was used to determine the differentially expressed proteins (DEPs) in T2 vs. T1 and T1 vs. T3 [70]. Protein functions were annotated against the databases including SwissProt and Gene Ontology (GO) [71,72].

### 4.7. RNA Extraction and qRT-PCR

Generally, gene expression can be applied to indirectly validate the protein expression [67,73,74]. In this study, qRT-PCR was used to identify the expression level for genes encoding the candidate proteins. Briefly, RNA samples were extracted from roots using a plant RNA kit. Primer sequences (Table 2) were designed in primer BLAST NCBI. RNA samples were extracted from the roots using a plant RNA kit (R6827, Omega Bio-Tek, Inc., Norcross, GA, USA) according to the manufacturer’s protocol. First-strand cDNA was synthesized using a FastKing RT Kit. PCR amplification was carried out using a SuperReal PreMix. *Actin* was used as an internal reference and the relative expression level (REL) was calculated using a 2^−^^△△Ct^ method [75].

### 4.8. Statistical Analysis 

All the measurements were performed using nine biological replicates and three technical replicates. Statistical analysis was performed using SPSS 22.0. One-way analysis of variance and Duncan multiple comparison tests were performed, with *p* < 0.05 as the basis for significant differences.

## 5. Conclusions

From the above observations, the root activity of cold-resistant *Vitis* species is greater than that of cold-sensitive species, and it declines with the decrease in temperature. The DEPs observed by proteomic analysis suggest that there was a significant difference in protein expression between the cold-resistant and cold-sensitive species in response to low temperature. The biological function of the 25 DEPs involved in stress response, bio-signaling, metabolism, energy, and translation should be further investigated using transgenic assays.

## Figures and Tables

**Figure 1 plants-11-00971-f001:**
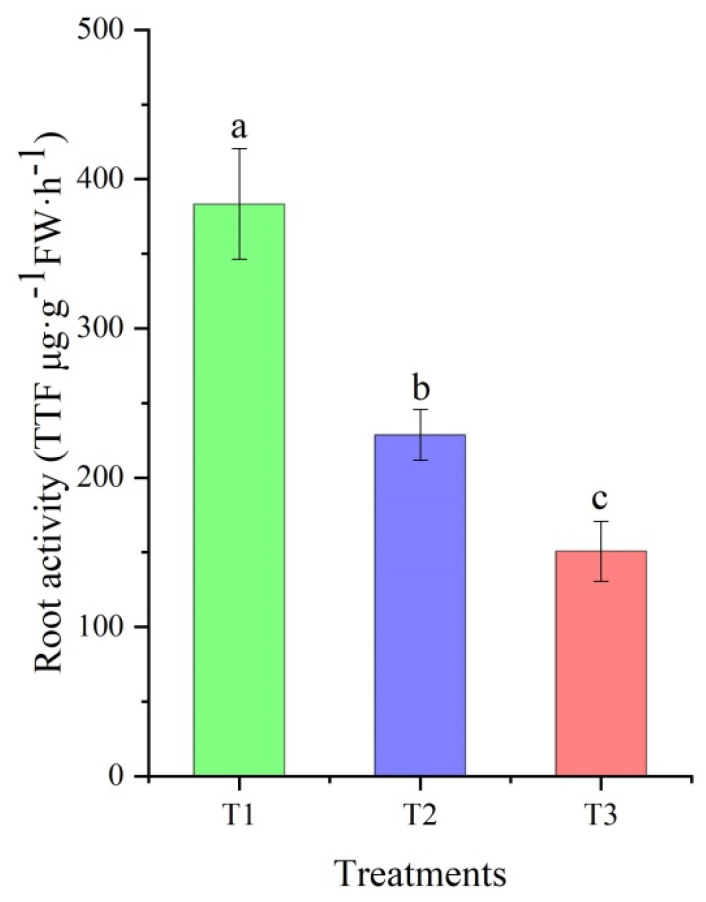
Differences in root activity in *Vitis riparia* × *Vitis labrusca* and Cabernet Sauvignon with different treatments. T1: *V**. riparia* × *V**. labrusca* at −4 °C; T2: *V**. riparia* × *V**. labrusca* at −15 °C; T3: Cabernet Sauvignon at −4 °C. Different lowercase letters represent a significant difference (*p* < 0.05) for the different treatments.

**Figure 2 plants-11-00971-f002:**
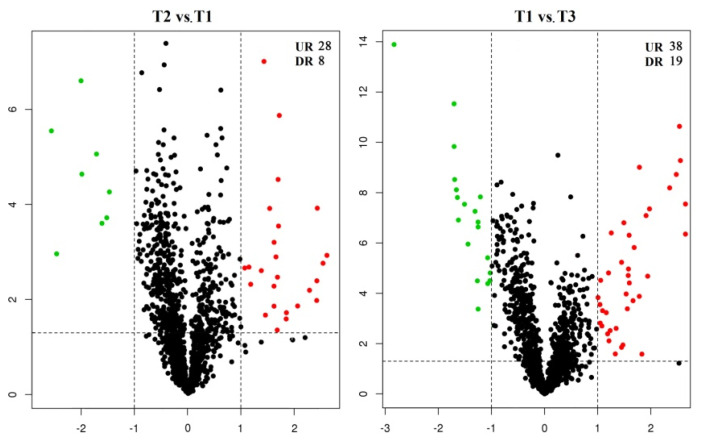
Volcano plot for roots of *Vitis riparia* × *Vitis labrusca* and Cabernet Sauvignon with different treatments. T1: *V. riparia* × *V. labrusca* at −4 °C; T2: *V. riparia* × *V. labrusca* at −15 °C; T3: Cabernet Sauvignon at −4 °C. UR, upregulated; DR, downregulated.

**Figure 3 plants-11-00971-f003:**
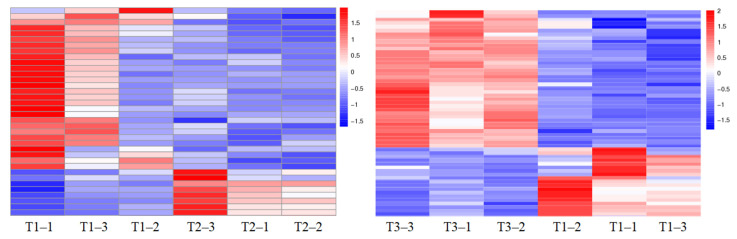
Cluster heat maps of the differentially expressed proteins (DEPs) in T1, T2, and T3 treatments. The values > 0 in the images indicate UR, while the values < 0 indicate DR. The order of the DEPS in the images was shown in Appendix A.

**Figure 4 plants-11-00971-f004:**
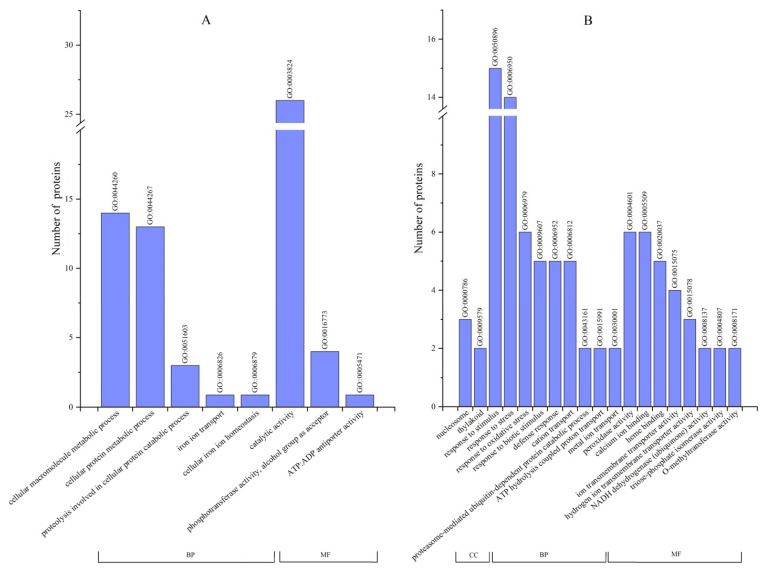
Functional classification of DEPs based on GO database for different treatments. (**A**): T2 vs. T1, (**B**): T1 vs. T3. CC, cellular component; BP, biological process; MF, molecular function.

**Figure 5 plants-11-00971-f005:**
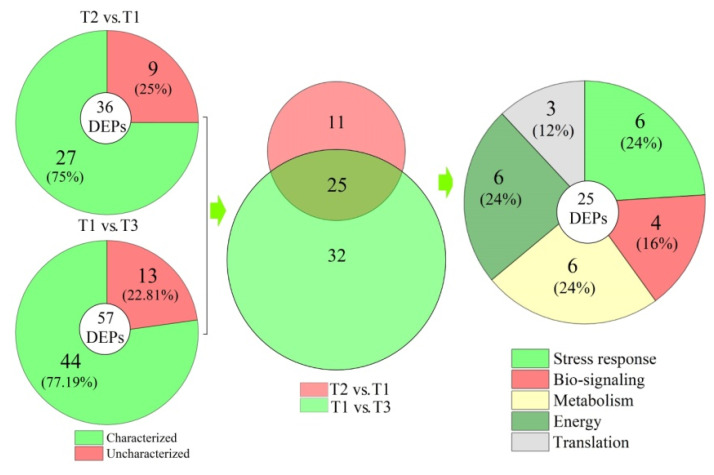
Distribution and classification of DEPs in roots of *V. riparia* × *V. labrusca* and Cabernet Sauvignon with different treatments.

**Figure 6 plants-11-00971-f006:**
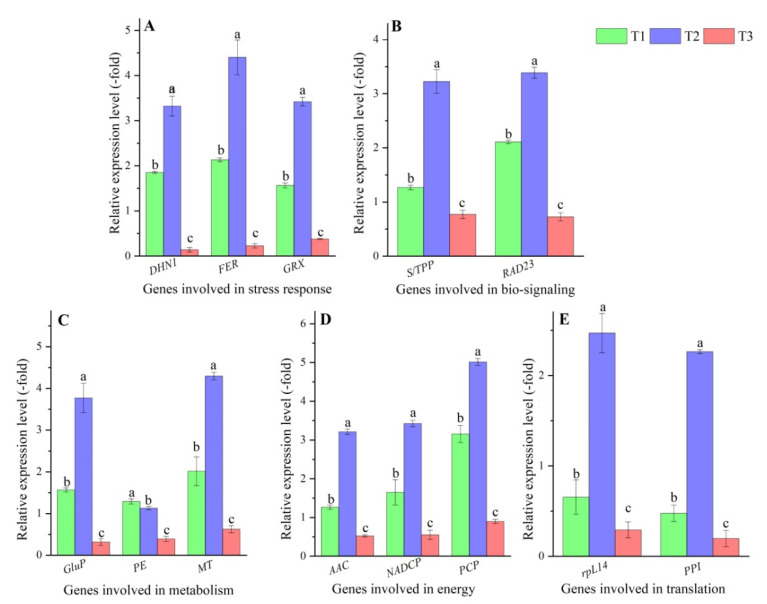
The relative expression level of genes involved in stress response (**A**), bio-signaling (**B**), metabolism (**C**), energy (**D**), and translation (**E**) at different treatments, determined by qRT-PCR. Different lowercase letters represent a significant difference (*p* < 0.05) in the different treatments for the same gene.

**Table 1 plants-11-00971-t001:** The 25 DEPs identified in *V. riparia* × *V. labrusca* and Cabernet Sauvignon with different treatments.

Protein Name (Abbreviation)	SwissProt ID	log_2_ FC (T2 vs. T1)	log_2_ FC (T1 vs. T3)
**Stress response (6)**			
Dehydrin (DHN1)	Q4VT48	1.34	1.78
SHSP domain-containing protein (SHSPCP)	F6HJZ4	1.76	2.01
Usp domain-containing protein (USPCP)	F6H727	1.60	2.69
Ferritin (FER)	A5BV73	2.83	1.83
Glutaredoxin-dependent peroxiredoxin (GluDP)	A5ARL2	2.72	1.92
Glutathione peroxidase (GPX)	D7TW03	1.28	1.73
**Bio-signaling (4)**			
Protein kinase domain-containing protein (PKCP)	A5ALY7	1.44	1.27
Serine/threonine-protein phosphatase (S/TPP)	D7TV73	1.98	2.92
Non-specific serine/threonine protein kinase (nsS/TPK)	F6H1V3	1.75	2.07
Ubiquitin receptor RAD23 (RAD23)	D7T959	1.22	1.35
**Metabolism (6)**			
Alpha-1,4 glucan phosphorylase (GluP)	D7SXJ4	2.65	1.36
1,4-alpha-glucan branching enzyme (GluBE)	E0CQR2	0.58	1.57
Pectinesterase (PE)	F6HZ64	0.63	2.92
Abhydrolase 3 domain-containing protein (ABHD3CP)	F6HQC6	0.84	1.21
Proline iminopeptidase (ProIP)	D7T3J3	2.28	1.94
Methyltransferase (MT)	A5B620	2.61	1.75
**Energy (6)**			
ADP/ATP carrier protein (AAC)	A5BVR2	1.63	1.61
AAA domain-containing protein (AAACP)	D7TZI9	1.61	1.59
NAD(P)-bd dom domain-containing protein (NADCP)	F6HL96	1.59	2.06
NADH dehydrogenase [ubiquinone] 1 beta subcomplex subunit 7 (NDUFB7)	A5BAM7	1.82	2.61
Phytocyanin domain-containing protein (PCP)	A5C3C3	1.85	2.83
Succinate dehydrogenase [ubiquinone] flavoprotein subunit (SDHFS)	A5BGN3	1.69	1.88
**Translation** **(3)**			
Ribosomal protein L14 (rpL14)	B6VJZ7	2.61	2.80
40S ribosomal protein S21 (rpS21)	A5BUA6	2.76	1.67
Peptidylprolyl isomerase (PPI)	D7UDY0	1.61	2.51

**Table 2 plants-11-00971-t002:** Sequences of primers used in qRT-PCR analysis.

Protein Name (Abbreviation)	Sequences (5′ to 3′)	Amplicon Size (bp)	Accession No.
Actin	Forward: CGCAGAGCACTTCTTTCCCA	181	XM_010657947.2
	Reverse: ATAGTGATGCCGCCTGATCC		
Dehydrin (DHN1)	Forward: ACCCAGTCCATCAAACCGAG	113	NM_001281292.1
	Reverse: GGATGAAGAGCTGCCGGATT		
Ferritin (FER)	Forward: GGAGCAGGACCAAGACCAAG	138	AM472371.2
	Reverse: GGAGATGGTGGGAAGCTCTG		
Glutathione peroxidase (GPX)	Forward: CACCGTTAAGGATGCTGAGG	153	XM_002272900.4
	Reverse: GGCCTTGATCTTTGTACTTCTCG		
Serine/threonine-protein phosphatase (S/TPP)	Forward: TCAACTGCCTTCCTGTAGCC	122	XM_002277780.3
	Reverse: TGGTACATCAACAGGGCGAG		
Ubiquitin receptor RAD23 (RAD23)	Forward: CAATGGGTTTTGACCGTGCC	170	XM_002282316.3
	Reverse: TGGTTCTAGGGGGATGGAGG		
Alpha-1,4 glucan phosphorylase (GluP)	Forward: GAGGCTTTGCGTGAACTTGG	105	XM_002279039.3
	Reverse: CAGAAAGCAGGAAGCAAGCC		
Pectinesterase (PE)	Forward: TGCTGATGTTGGTGGGAGAC	173	XM_002271629.4
	Reverse: CACTGCTTGGTGATTGCTCG		
Methyltransferase (MT)	Forward: TAGGCGTGAGATGTGTGTGG	197	AM447844.2
	Reverse: GACCTGCCTGCTTCGGTAAG		
ADP/ATP carrier protein (AAC)	Forward: CCCTTGGGGCTTTTTCCCAT	160	AM472940.2
	Reverse: GGGCAAAGCATGTCCACTAC		
NAD(P)-bd dom domain-containing protein (NADCP)	Forward: TGGTTGGGTCTATGGGAGGA	174	XM_010655958.2
	Reverse: GTAATTCCCGGATGCCACCT		
Phytocyanin domain-containing protein (PCP)	Forward: GCCCAGACCATTACGGATAGG	182	AM480712.2
	Reverse: CCACATTGGTCGGCTTTGAG		
Ribosomal protein L14 (rpL14)	Forward: CCGCGACTTCGGTCTTTTTC	134	FN595512.1
	Reverse: GCCTTACGTCTGTCTGGAGG		
Peptidylprolyl isomerase (PPI)	Forward: TCGGGGGAAACTCACAGATG	141	XM_002271020.4
	Reverse: TTTCGCTTCTCACCCACACA		

## Data Availability

The datasets are publicly available at ProteomeXchange with the dataset identifiers PXD009435 and PXD009837.

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
