# Peer review of "Comparative Proteomics Reveals the Difference in Root Cold Resistance between *Vitis. riparia* × *V. labrusca* and Cabernet Sauvignon in Response to Freezing Temperature"

_plants, 2022, doi:10.3390/plants11070971_

Round 1

Reviewer 1 Report

In the ms entitled "Comparative Proteomics Reveals the Difference of Root Cold Resistance between Vitis. riparia × V. labrusca and Cabernet Sauvignon in Response to Freezing Temperature", the authors use proteomic and transcriptomic methods to compare the cold resistance between two grapevine cultivars. Although the protein work is very interesting, I believe that the ms must be severely improved in order to be more comprehensive. Therefore, I have the following questions/suggestions to the authors:

  • considering that the x-axis in the volcano plot represents log(2)FC, I wonder why do you use the value 0.5 (or -0.5) to determine a protein as DEP. As far as I know, DEPs (or DEGs in the case of RNAseq) should be at least 2-fold over- or under-expressed between samples i.e., -1<log(2)FC<1. I have the same point considering Table 1.
  • you should explain the heatmaps and the clustering in Fig. 3 i.e. which proteins are UR or DR, their function etc. 
  • instead of presenting the qPCR results as T2 vs T1 and T1 vs T3, you should provide separetely T1, T2 and T3 values or normalize them to the lower value. That way it will be more easy to understand the fold change between the samples.  
  • please explain better (i.e. in 2-3 sentences) the meaning of root activity to make it more compehensive
  • you should combine sections 2.3 - 2.7 in one section. Maybe it would also be better to present the qPCR results in another section. 
  • in the Discussion section, please do not use "For the DEPs involved in..." at the beginning of each paragraph. Just write a sentence about the protein(s) or the fuction(s) they are related to. 
  • you should also elaborate the gene expression results in Discussion.
  • you should explain the "translation-based regulation of roots... " you state in Conclusion section. 
  • you should provide the location of the experimental vineyard in Materials and Methods. 
  • if you performed SDS-PAGE and Western blotting you should provide at least a representative picture. Otherwise, there is no reason to mention it. 
  • you should provide two Tables (as Supplementary material) of the 56 and 143 proteins identified as DEPs (like you did in Table 1 for the 25 common proteins). 
  • the suppl. figures (S1, S2 and S3) are not referred in the manuscript. Additionally, they need more descriptive legeds (e.g. what are the green boxes in S3, DEPs not DAPs, which are UR or DR). 
  • finally I suggest to use "grapevine" and "literature" instead of "grapevines" and "literatures" in the text.  

Author Response

Many thanks for your and reviewer’s comments that are helpful to improve our paper much better now. We have tried to address and correct each comment. Reviewers’ comments are attached below as well as our responses shown in bold. Revised parts (descriptions) are highlighted in red in the manuscript. 

The specific revisions have been uploaded in the PDF document.

Reviewer 2 Report

  1. Please consider having an individual or service well versed in English for scientific manuscripts vet this manuscript.
  2. Page 9, Line 264: insufficient details. What kind of soil?  Number of vines per cv.?  How was soil water content monitored and controlled?  Glass separations imply a trench was dug, glass separations placed, and soil replaced or were glass separations somehow pushed into the soil?
  3. Page 9, line 265: most cultivars (e.g. Cabernet Sauvignon) are scions which are grafted onto rootstock. The description indicates these vines are own-rooted, rather than grafted.  The authors need to discuss the relevance of this research to the "real world" in which rootstocks generally confer biotic and abiotic stress resistance/ tolerance or desirable scion growth characteristics. 
  4. Page 9, line 268: how is this known?
  5. Page 9, line 269: how was this monitored and controlled?
  6. Page 9, line 270: so, nine biological replicates?
  7. Page 9, line 271: any soil temperature data at 20 cm depth? This could have been collected with a thermocouple and digital thermometer.  This is critical as it is highly unlikely that soil temperature at 20 cm depth and at the soil surface are equal.
  8. Page 9, line 272: how were the roots collected? It is assumed that the soil was rinsed off.  If so, what was the water temperature? Could the roots have been altered in their physiological state by water at a different temperature?
  9. Page 9, line 273: no root collection at a soil surface temperature of -15 C? Why?
  10. Page 9, line 276: The citation is not readily available. What is meant by root activity?  These were collected at 20 cm depth?  Once again, the authors need to detail how the roots were removed from the soil, as root breakage, especially root tip breakage, is common during such studies.
  11. Page 9, line 287: doesn’t TCA/ acetone precipitate proteins? The proteins need to be dissolved in some solvent first.  This step is lacking.
  12. Page 9, line 294: how? SDS-PAGE is more qualitative that quantitative.  Some method such as the Bradford assay (for example) is needed for quantitative data.
  13. Page 10, line 307: something is missing.
  14. Page 10, lines 316-318: this sentence makes no sense. Moreover, the authors need to describe the primary and secondary antibodies used and sources.
  15. Page 10, line 321: insufficient information on RNA extraction.
  16. Page 10, lines 323-324: there are typically many actin genes. Which one was used?  Was the stability of this actin gene assessed over time/ treatment so that it provided a stable endogenous reference? If so, how? Lastly, best practice is to use at least endogenous reference genes.  Was this done? 
  17. Page 10, line 326: how many technical replicates?

Author Response

(The authors gave the same response as above.)

Reviewer 3 Report

A very interesting work that contributes to the knowledge of the grapevine's resistance to cold.

Author Response

Thanks very much for your affirmation and approval to our manuscript.

Round 2

Reviewer 1 Report

The new version of the ms is obviously improved and the authors responsed to many of my suggestions. However, there are still two points to be addressed: 

  • first, I still believe that the authors should use -1<logFC<1 in their analysis of protein expression in their treatments. Indeed, low abudance proteins may play crucial role in plant development, but this is not the point when comparing different samples - they can have a protein with low (or high) abudance in both samples and they want to know the relative abudance (further, there is no sample comparison nor any clue for using |logFC|>0.5 in the new references they provide). I think that it will better to keep about 25 proteins in T1 vs T2 and about 60 proteins in T1 vs T3 and to be sure that they are really differentially expressed (i.e. at leats 2-fold change Up or Down). Additionally, all of their common DEPs are |logFC|>1, and most of the 0.5<|logFC|<1 are uncharacterized proteins, according to the Tables.
  • second, the authors have to correct the RT-qPCR results presented. As they use the 2^ΔCt method i.e., 2^[Ct(reference gene) - Ct (target gene)],  it is not possible to have values < 0    

Author Response

The new version of the ms is obviously improved and the authors responsed to many of my suggestions. However, there are still two points to be addressed: 

  • First, I still believe that the authors should use -1<logFC<1 in their analysis of protein expression in their treatments. Indeed, low abundance proteins may play crucial role in plant development, but this is not the point when comparing different samples - they can have a protein with low (or high) abundance in both samples and they want to know the relative abundance (further, there is no sample comparison nor any clue for using |logFC|>0.5 in the new references they provide). I think that it will better to keep about 25 proteins in T1 vs T2 and about 60 proteins in T1 vs T3 and to be sure that they are really differentially expressed (i.e. at least2-fold change Up or Down). Additionally, all of their common DEPs are |logFC|>1, and most of the 0.5<|logFC|<1 are uncharacterized proteins, according to the Tables.

Thanks very much for your suggestion, the criterion of “|log2(fold-change)| ≥ 1” has replaced “|log2(fold-change)| ≥ 0.5” to determine the DEPs in T2 vs. T1 and T1 vs. T3.( Page 11, lines 345)

Indeed, among the 25 DEPs, although the values of three proteins (GluBE, PE and ABHD3CP) (Page 6, Table 1) showed 0.5<|logFC|<1 in T2 vs. T1, while they showed |logFC|>1 in T1 vs. T3. Thus, the 25 proteins are still kept in this present study.

  • Second, the authors have to correct the RT-qPCR results presented. As they use the 2^ΔCt method i.e., 2^[Ct(reference gene)-Ct (target gene)], it is not possible to have values < 0.      

Thanks for your kind reminder, the qRT-PCR results have been corrected in Figure 6, and some error values have also been revised in the text. (Page 7; lines 150-164)

Reviewer 2 Report

Comments to the authors (Cover Sheet)

  1. Regarding the authors’ response to my point #2, this Supplemental figure does not appear in the latest supplemental figure file. Moreover, the cover letter refers to this figure as S1, while the figure legend clearly indicates S2.  Lastly, if a thermometer was used, where are the data?  My concern still stands - soil temperature 20 cm below soil level is unlikely to reflect temperatures at the soil surface level. 
  2. Regarding the authors’ response to my point #7, again, where are the soil temperature data from this thermometer?
  3. Regarding the authors’ response to my point #9, the authors miss the point of the question. The authors had two sampling temperatures for the hybrid, -4 °C and -15 °C.  Cabernet Sauvignon had only one sampling temperature, -4 °C.  Why didn’t the authors sample at -15 °C?
  4. Regarding the authors’ response to my point #10, their elucidation is still confusing and somewhat contradictory. Does their methodology imply that the liquid nitrogen boiled off and the root tips picked out of the soil because they were flexible, i.e., thawed?  If the root/ soil mix was frozen, then allowed to thaw (resulting in flexible roots), wouldn't this compromise the resulting protein and RNA data due to disruption of cellular compartments and mixing of proteases with the proteins and RNAses with the RNA?  Or did the authors somehow pick flexible roots out of a mix of boiling liquid nitrogen and soil?
  5. Regarding the author’s response to my point #12, there is still no indication as to whether proteins were loaded on an equal basis or what quantity was loaded.
  6. Regarding the authors’ response to my point #16, the authors still miss the point of the question. Melt curves are useful, but a method/ software such as NormFinder should be used to ascertain the stability of an endogenous reference gene between treatments or over time.  Moreover, current standards for such experiments require the use of two or more endogenous reference genes to assure that the data are reliable.  Were two or more genes used?

Comments to the authors on the revised manuscript

  1. Introduction: what is the significance of cyclooxygenases?
  2. Page 2: please explain how this was "spatio-temporally". An indication of the length of time between -4° C and -15 °C collection points might help.
  3. Page 2, Results 2.1: the authors should mention how triphenylformazan relates to activity. Secondly, please confirm that the T1 root activity was 2.43 FOLD greater than that of T3 and not 2.43 TIMES greater.  These are mathematically distinct, but unfortunately used interchangeably by too many authors.  This question reoccurs many times throughout the manuscript. 
  4. Page 3, Results 2.2: Perhaps the authors meant “compared to” rather than “enriched”.
  5. Page 3, Results 2.2: perhaps the authors meant "...31 enriched proteins were sorted into molecular functions,..."?
  6. Table 1: is DHN1 taken from a classification of Vitus dehydrins or did the authors assign this number?
  7. Please consider using “greater than” rather than “stronger” when comparing values.
  8. Discussion, first paragraph, last sentence: consider “up-regulated” rather than “over-expression”. “Over-expression” usually connotes a transgenic plant that over-expresses a transgene.  This usage occurs several more times and should be corrected.
  9. Page 8, first paragraph, last sentence: this sentence is confusing.
  10. Page 8, second paragraph, first sentence: A DEP could be differentially expressed higher or lower. The sentence seems to imply that all/ most DEPs are involved in signaling.  Please consider rewriting this sentence. 
  11. Page 8, second paragraph, last sentence: there was only one timepoint at each temperature. This is sheer conjecture.
  12. Page 8, third paragraph, middle: what is meant by "almost"? Either they were or they weren't differentially expressed by your pre-established criteria. 
  13. Page 8, third paragraph, second to last sentence: there is something missing from this sentence.
  14. Page 8, third paragraph, last sentence: speculation. The authors don't have any data to support this contention.
  15. Page 8, Section 4.1: the authors still don't indicate HOW the soil water content was controlled, only how it was monitored.
  16. Page 10, Section 4.3: the authors have not corrected error that the acetone precipitates, not dissolves proteins.

Author Response

Comments to the authors (Cover Sheet)

1. Regarding the authors’ response to my point #2, this Supplemental figure does not appear in the latest supplemental figure file. Moreover, the cover letter refers to this figure as S1, while the figure legend clearly indicates S2.  Lastly, if a thermometer was used, where are the data?  My concern still stands - soil temperature 20 cm below soil level is unlikely to reflect temperatures at the soil surface level. 

Thanks very much for your comments. Firstly, we are sorry for forgetting to submit the latest Supplementary Materials at the Round 1 response. After the Associate Editor reminder, the latest Supplementary Materials containing Figure S1 and Figure S2 have been re-submitted.

For the measure of soil temperature, the method of a thermometer buried into the soil at the depth 20 cm was used. Generally, most of the roots of Vitis, especially in lateral roots, mainly distribute 20 cm below soil; thus, we measure the 20 cm soil temperature and also collect the roots at 20 cm.

2.Regarding the authors’ response to my point #7, again, where are the soil temperature data from this thermometer?

According to your comments, the data of 20 cm soil temperatures have been supplemented as Figure S2 in the Supplementary Materials. (Page 12, lines 377-378)

Figure S2: Changes of average temperatures at the 20 cm depth soil in December and January at the experiment site.  

3. Regarding the authors’ response to my point #9, the authors miss the point of the question. The authors had two sampling temperatures for the hybrid, -4 °C and -15 °C.  Cabernet Sauvignon had only one sampling temperature, -4 °C.  Why didn’t the authors sample at -15°C?

Firstly, in order to reveal the difference in varieties, two varieties include Vitis. riparia × V. labrusca and Cabernet Sauvignon were selected because the former is cold-resistant and the later is cold-sensitive.

Secondly, in order to reveal the difference in temperatures, the cold-resistant variety (Vitis. riparia × V. labrusca) was used. Thus, the cold-sensitive variety (Cabernet Sauvignon) was not determined.

4. Regarding the authors’ response to my point #10, their elucidation is still confusing and somewhat contradictory. Does their methodology imply that the liquid nitrogen boiled off and the root tips picked out of the soil because they were flexible, i.e., thawed?  If the root/ soil mix was frozen, then allowed to thaw (resulting in flexible roots), wouldn't this compromise the resulting protein and RNA data due to disruption of cellular compartments and mixing of proteases with the proteins and RNAses with the RNA?  Or did the authors somehow pick flexible roots out of a mix of boiling liquid nitrogen and soil?

Indeed, it is difficult to separate the roots from the freezing soil. To our knowledge, we have no better way to collect the roots with the method “by using the liquid nitrogen boiled off and then immediately picked the root tips out of the soil”. After all, it is significant different in physical characteristics (e.g. structures and compositions) between the roots and the soil.  

5. Regarding the author’s response to my point #12, there is still no indication as to whether proteins were loaded on an equal basis or what quantity was loaded.

In this study, the equal amount of proteins were used for iTRAQ labeled. The information: “After protein quantification, the equal amount of proteins (150 μg)---” has been added in the text. (Page 10, lines 321-322)

6. Regarding the authors’ response to my point #16, the authors still miss the point of the question. Melt curves are useful, but a method/ software such as NormFinder should be used to ascertain the stability of an endogenous reference gene between treatments or over time.  Moreover, current standards for such experiments require the use of two or more endogenous reference genes to assure that the data are reliable.  Were two or more genes used?

In this study, only the actin gene was used for the reference gene; first, the actin gene has been investigated before the quantification of gene expression and found to be stability based on the Melt curves generated from the qRT-PCR instrument (ABI QuantStudio 5).  

Comments to the authors on the revised manuscript

1. Introduction: what is the significance of cyclooxygenases?

According to your comments, the description has been revised to: “cardiovascular benefits, and cancer chemopreventive activity”. (Page 1, lines 33-34)

 2. Page 2: please explain how this was "spatio-temporally". An indication of the length of time between -4° C and -15 °C collection points might help.

The length “after 30 d of T1” of time between -4° C and -15 °C has been added in the text. (Page 2, lines 66-67)

3. Page 2, Results 2.1: the authors should mention how triphenylformazan relates to activity. Secondly, please confirm that the T1 root activity was 2.43 FOLD greater than that of T3 and not 2.43 TIMES greater.  These are mathematically distinct, but unfortunately used interchangeably by too many authors. This question reoccurs many times throughout the manuscript. 

According to your comments, the role of triphenylformazan has been mentioned in the text: “After the roots measured by the triphenyl tetrazolium chloride (TTC) method, the root activity was evaluated based on the amount of reactant triphenylformazan.”. (Page 2, lines 74-75).

Indeed, it is the “times greater ”, which has been added in the text. (Page 2, line 77)

4. Page 3, Results 2.2: Perhaps the authors meant “compared to” rather than “enriched”.

The words “compared to” have replaced “enriched against”. (Page 3, line 92)

5. Page 3, Results 2.2: perhaps the authors meant "...31 enriched proteins were sorted into molecular functions,..."?

The word “sorted” has replaced “enriched” throughout the text. (Page 3, lines 93, 97, 100, 106 and 108)

6. Table 1: is DHN1 taken from a classification of Vitis dehydrins or did the authors assign this number?

Yes, the DHN1 is taken from the Vitis. The DHN1 is abbreviation of dehydrin, which is shown in the SwissProt database: https://www.uniprot.org/uniprot/Q4VT48.

7. Please consider using “greater than” rather than “stronger” when comparing values.

The description of “greater than” has replaced “stronger than” throughout the text.

8. Discussion, first paragraph, last sentence: consider “up-regulated” rather than “over-expression”. “Over-expression” usually connotes a transgenic plant that over-expresses a transgene.  This usage occurs several more times and should be corrected.

Thanks for your reminder, the “up-regulation” has replaced “over-expression” throughout the text.

9. Page 8, first paragraph, last sentence: this sentence is confusing.

The sentence has been revised: “Meanwhile, the up-regulation of the DEPs was observed in cold-resistant V. riparia × V. labrusca compared with cold-sentenisve Cabernet Sauvignon.” (Page 8, lines 182-184)

10. Page 8, second paragraph, first sentence: A DEP could be differentially expressed higher or lower. The sentence seems to imply that all/ most DEPs are involved in signaling.  Please consider rewriting this sentence. 

According to your comments, the sentence has been revised: “Up- or down-regulated DEPs related to bio-signaling can perceive the cold stress and transfer it to the cellular response.” (Page 8, line 214)

11. Page 8, second paragraph, last sentence: there was only one time point at each temperature. This is sheer conjecture.

Indeed, this is speculation based on the results. Thus, the sentence has been revised: “The up-regulation of these proteins maybe constantly on the alert to ensure that plants are not injured by low temperatures .” (Page 8, line 228)

12. Page 8, third paragraph, middle: what is meant by "almost"? Either they were or they weren't differentially expressed by your pre-established criteria. 

The word “almost” has been deleted. (Page 9, line 240)

13. Page 8, third paragraph, second to last sentence: there is something missing from this sentence.

After checking the sentences, we are sure that all the six identified DEPs have been mentioned in the text.

14. Page 8, third paragraph, last sentence: speculation. The authors don't have any data to support this contention.

Indeed, the sentence: The down-regulation at −15°C might be a part of the mechanism associated with the delay of senescence and death [49].” is speculated based on our data and previous literatures; after all, these DEPs related to metabolism need further investigation.

15. Page 8, Section 4.1: the authors still don't indicate HOW the soil water content was controlled, only how it was monitored.

Thanks for your reminder. In this study, “mulching film” is used to cover the surface of the soil. Generally, during overwintering stage, the field is not irrigated in open field; on the one hand, less water is evaporated from the frozen soil; on the other hand, abundance water has been irrigated before the coming of the winter. (Page 10, line 287)

16. Page 10, Section 4.3: the authors have not corrected error that the acetone precipitates, not dissolves proteins.

The error “then dissolved in pre-chilled (−20°C) trichloroacetic acid/acetone” has revised to: “then added in pre-chilled (−20°C) trichloroacetic acid/acetone”. (Page 10, line 312)

Round 3

Reviewer 1 Report

The authors should correct Figure 2 (volcano plots) and Figure 5 (venn diagrams) as well as the Supplementary Tables 1 and 2, considering that they now use |logFC| >1 to determine the differential expressed proteins.

Further, they should replace the number of proteins (DEPs) in the text (e.g. in Line 88 "A total of 56 and 143 DEPs were obtained..." and also thereafter) with the correct numbers. 

Finally, I would appreciate the authors revising carefully the manuscript, the figures and the suppl. material and  making all the above corrections in one round of revision. 

Author Response

Dear Reviewer,

Thanks very much for your suggestion and comments that are helpful to improve our manuscript.  Reviewers’ comments are attached below as well as our responses shown in bold. Revised parts (descriptions) are highlighted in red in the manuscript.

Comments from the reviewer:

The authors should correct Figure 2 (volcano plots) and Figure 5 (venn diagrams) as well as the Supplementary Tables 1 and 2, considering that they now use |logFC| >1 to determine the differential expressed proteins.

The Figure 2 (volcano plots), Figure 2 (heat map) and Figure 5 (venn diagrams) as well as the Supplementary Tables 1 and 2 have been corrected (Page 3, line 111; Page 4, line 116; Page 5, line 130).

Further, they should replace the number of proteins (DEPs) in the text (e.g. in Line 88 "A total of 56 and 143 DEPs were obtained..." and also thereafter) with the correct numbers. 

The number of proteins (DEPs) have been corrected to 36 and 57 througout the text (Page 3, line 87; Page 4, line 126; Page 11, lines 373-374).

Finally, I would appreciate the authors revising carefully the manuscript, the figures and the suppl. material and  making all the above corrections in one round of revision. 

Reviewer 2 Report

The authors have addressed my concerns and the manuscript is significantly better.  

Author Response

Thanks very much for your kind review and comments.

Round 4

Reviewer 1 Report

The authors have made all the corrections in the ms, therefore I believe that now can be accepted for publication.